# Strategies and Trends in COVID-19 Vaccination Delivery: What We Learn and What We May Use for the Future

**DOI:** 10.3390/vaccines11091496

**Published:** 2023-09-16

**Authors:** Giuseppe Tradigo, Jayanta Kumar Das, Patrizia Vizza, Swarup Roy, Pietro Hiram Guzzi, Pierangelo Veltri

**Affiliations:** 1Department of Computer Science, eCampus University, 22060 Novedrate, Italy; giuseppe.tradigo@uniecampus.it; 2Longitudinal Studies Section, Translation Gerontology Branch, National Institute on Aging, National Institutes of Health, Baltimore, MD 21224, USA; dasjayantakumar89@gmail.com; 3Department of Surgical and Medical Science, Magna Græcia University, 88100 Catanzaro, Italy; hguzzi@unicz.it; 4Network Reconstruction & Analysis (NetRA) Lab, Department of Computer Applications, Sikkim University, Gangtok 737102, India; sroy01@cus.ac.in; 5Department of Computer Science, Modelling, Electronics and Systems, University of Calabria, 87036 Rende, Italy; pierangelo.veltri@unical.it

**Keywords:** vaccination strategy, COVID-19, epidemiology, vaccine distributions

## Abstract

Vaccination has been the most effective way to control the outbreak of the COVID-19 pandemic. The numbers and types of vaccines have reached considerable proportions, even if the question of vaccine procedures and frequency still needs to be resolved. We have come to learn the necessity of defining vaccination distribution strategies with regard to COVID-19 that could be used for any future pandemics of similar gravity. In fact, vaccine monitoring implies the existence of a strategy that should be measurable in terms of input and output, based on a mathematical model, including death rates, the spread of infections, symptoms, hospitalization, and so on. This paper addresses the issue of vaccine diffusion and strategies for monitoring the pandemic. It provides a description of the importance and take up of vaccines and the links between procedures and the containment of COVID-19 variants, as well as the long-term effects. Finally, the paper focuses on the global scenario in a world undergoing profound social and political change, with particular attention on current and future health provision. This contribution would represent an example of vaccination experiences, which can be useful in other pandemic or epidemiological contexts.

## 1. Introduction

Vaccination is the most effective method to contrast infectious diseases. The Expanded Program on Immunization (EPI) has been defined by the World Health Organization (WHO) in the early 1970s to provide large-scale access to vaccines for children. Nevertheless, approximately 20 million infants worldwide still do not have access to immunization services. The WHO, on March 2020, declared that the coronavirus had become a pandemic [1] since the containment measures could not stop the spread. Research focused on the development of novel vaccines to prevent the spreading of the virus, while in parallel, some drugs were used to treat the illness [2,3,4,5].

Many primary pharmaceutical industries, supported by an unprecedented effort by governments, focused on defining novel vaccines. With surprising speed, the Food and Drug Administration (FDA) and European Medicine Agency (EMA) approved many candidate vaccines.

The COVID-19 pandemic boosted the necessity of vaccinations and accelerated the design and testing of vaccines, using new procedures. We can consider that the vaccination campaign started at the early beginning of 2021 in some countries and then was extended to all other countries, signaling the decline in deaths and number of infections. We have since had an increasing number of vaccinations and increasingly efficient protocols. Many COVID-19 vaccines have regulatory approval, most of which have provisional authorization, motivated by the results of trials. Also, vaccines have been distinguished among young citizens (under 12 or 18) and adult ones. Table 1 reports an extracted number of vaccines; an additional and updated number of vaccines with an updated geographical distribution and diffusion by country can be found in [6]. The different authorization types required time (and in some cases delay). This is relevant to consider for the large-scale discussion and definition of vaccine application protocols. Indeed, COVID-19 variants have sometimes been faster with respect to the authorization protocol process in different countries. This important aspect should be considered when managing a huge scale and intercountry phenomena. However, not withstanding the rapid response to COVID-19, it became clear that it was impossible to satisfy the demand for vaccines and hence the need for prioritization strategies [7]. At least 72.3% of the world population received a single dose of the COVID-19 vaccine by the end of 2022. The source of the vaccination trend can be found on the official website in [8] and in many studies monitoring real-time data [9]. In lower-income countries, some populations are still receiving vaccines. At the same time, the number of COVID-19 variants has increased, and the number of infections is still affecting populations. During the evolution of the pandemic, variants appeared to be a relevant issue to be considered, both with respect to vaccination response and disease expression. Moreover, tracking variants has become a useful task for relating virus with vaccine efficacy [10,11]. Indeed, a new topic related to the long-term effects of vaccination on virus containment arose and is reported in [12]. This represents a valid and unique world health status that has to be considered.

In China, the number of positive cases started to boost, while the number of vaccinated people remained low by the middle of 2022 (see data on https://tradingeconomics.com/china/coronavirus-vaccination-total (accessed on 20 August 2023)). Thus, lockdown containment strategies are still the only tentative to contrast the diffusion. Vaccine strategy is a crucial challenge to fight COVID-19 and to minimize number of deaths and infections [13].

Here, we consider the main processes related to COVID-19 vaccine diffusion within the pandemic and different variants of the COVID-19 virus [14,15,16,17]. Also, we focus on monitoring different strategies implemented by governments, global associations, and pharmaceutical industries. In particular, we focus on vaccination strategies across countries and their effects on different populations, stimulating a discussion about their consequences on life quality. The target is to highlight strategies and behaviors regarding vaccination distribution and effects on the population that can be considered a relevant track that can be followed in future similar emergency pandemic phenomena. Moreover, the onset of variants and their correlation with vaccination effects can be considered as relevant aspects for future decision-making.

Existing surveys cover different aspects of the question. Some reviews focus on the experimental aspects of vaccine development and test [18,19]. For instance, [20] focuses on the experimental workflow for the production of vaccines. Similarly, [21] presents a critical review of the bioinformatics methods and tools used to support vaccine development. Information about vaccinations can be found in several papers such as [22,23,24,25,26,27,28,29,30,31,32,33]. Aspects of the vaccine uptake have been treated in [34], while in [35], the authors analyze the problem of vaccine delivery. We herein treat issues related to long-term strategies and their effects on the worldwide vaccination process. The latter has been the best contrast against COVID-19, but it has also changed many life-related strategies. For this study, we followed the PRISMA 2020 methodology [36]. The chosen databases for searching the reference papers are: PubMed, MDPI, Springer, ACM Digital Library, and Science Direct, from which we considered the papers resulting from the search keywords “covid”, “vaccination strategy”, “vaccines”, and “Sars-Cov-2”, selecting the most relevant among recent ones.

In the following, we enumerate some of the relevant aspects treated in this paper:Vaccination strategies: We consider the applied methodologies used in different countries, and we focus on the problem of having heterogeneous strategies;Vaccines distribution: We focus on the problem of dose delivery among different countries as well as its relation to economic disparity among countries; in the latter case, we focus on strategies of cooperation among different areas;Variants and vaccines: We focus on variants and on different strategies for virus containment applied in different geographic areas post-emergency.

Finally, this paper highlights the effects of vaccination strategies on a dynamically changing social and sanitary worldwide scenario. Such strategies affect the evolution of the pandemic in both the emergency and the effects of the post-pandemic phase, also impacting the long-term scenario.

## 2. Methods

Vaccines can be seen as a prevention therapy in pandemics [37]. Even if the vaccination process reached high percentage values, the COVID-19 global immunization still needed to be reached. This is because citizens with three vaccine doses were still being affected by the COVID-19 virus variant. The authors of [38] report, for instance, the situation for vaccine allocation priority in different countries, whereas [39] shows the condition in Italy at the end of the third vaccine. Older citizens in Italy, for instance, received their fourth vaccine dose in 2022. Vaccination strategies have to be monitored with respect to the pandemic evolution even with decreasing values of mortality. Moreover, there are other social and political issues which must be considered [40,41]. For instance, regarding COVID-19 testing procedures for citizens moving abroad, the increasing number of positive cases forced governments to define new strategies for border management.

We study the vaccination distribution process and the public reaction to its effects, as well as the modification of social strategies depending on the degree of virus presence.

There exists six types of vaccines:Inactivated microorganisms: Consisting of viruses or bacteria killed by physical or chemical means (used for typhus, cholera, whooping cough, polio Salk);Live and attenuated microorganisms: Made up of live viruses or bacteria capable of reproducing in the individual host and guarantee lasting immunity (used for polio Sabin, TB, measles, rubella, mumps, chickenpox);Fractions of microorganisms: “Split” vaccines consist of fragmented viruses but without the purification of protective antigens so that they do not cause adverse reactions;Purified microbial antigens: Bacterial or viral components are purified and, in some cases, conjugated to carrier molecules (used for meningococcus, Hib, pneumococcus);Anatoxins (or toxoids): They are toxins treated with formol to obtain an antigenically intact product without toxicity;Vaccines from genetic manipulation: Produced with a part of the virus.

Another widely adopted classification for COVID-19 vaccines is as follows:mRNA vaccine, which is a type of vaccine containing cellular instructions for the production of S protein, a molecule found on the surface of the virus, which helps in creating antibodies [42,43,44]. These will be a defence layer in case of future infection by the COVID-19 virus. This type of virus does not interfere with DNA information in the cell nucleus. Examples in this category are Pfizer-BioNTech and Moderna COVID-19 vaccines;Vector vaccine is a type of vaccine using a modified virus (the viral vector) in which parts from the COVID-19 virus are inserted. The viral vector is acquired by the cells and causes them to produce COVID-19 S protein copies, which then induce the immune system to respond. Examples in this category are Janssen/Johnson & Johnson COVID-19 and AstraZeneca vaccines;Protein subunit vaccine includes parts of the virus that stimulate the immune system by using harmless S proteins.

The SARS-CoV-2 is a single strand of RNA made up of 30,000 building blocks wrapped in a protein envelope. mRNA vaccines instruct the immune system to recognize the pathogen. Synthetic mRNA molecules can induce the formation of a spike protein, which is sufficient to create an immune response and stimulate the production of antibodies. In other cases, vaccines are built from an adenovirus, whose genetic information is changed to convey parts of the new coronavirus suitable for stimulating an immune response.

The current landscape of existing vaccines is reported in Table 1 (data taken from [45]). Nevertheless, since the scenario is changing, updated vaccine trackers are available, such as the one provided by the London School of Hygiene and Tropical Medicine in [46].

The design of an effective vaccination strategy requires the availability of a computational model for designing and implementing interventions at all levels and ensuring the targeting of the desired goals. There exist some mathematical models to simulate the course of the disease and the effect of different strategies (e.g., containment, vaccination) [30]: forecasting based on time-series data; compartmental models; graph-based models.

To track the spread of infectious diseases, it is possible to use compartmental models. The whole population is assigned to compartments identified by labels such as: (1) Susceptible Infected (SIS); or (2) Susceptible, Infectious, and Recovered (SIR); or (3) Susceptible, Infectious, Recovered, and Vaccinated (SIRV). These models use mathematical functions, such as ordinary differential equations, to study the evolution of the population belonging to each compartment, e.g., in contact-based models, the population is modeled as a temporal graph G. Nodes of the graph represent the patients, while node labels represent the status of the individual obtaining one of the following values: Susceptible, Recovered, or Vaccinated. Temporal edges represent contacts among them, and the spread is easily represented by a Markovian model [31,47,48].

The vaccination campaign covers every population throughout the world [9,49]. The production rates of vaccines remained insufficient for several months to cope with the growing transmission rates of the emerging variants [13,50]. Consequently, there is the need to define and implement a prioritization strategy [51].

Vaccination strategies which started in December 2020 [52] report that by January 2021, 30 European countries had started the vaccination process and the vaccination tracking can be found in [53]. India launched its vaccination program for COVID-19 on 16 January 2021. By September 2021, every country had begun vaccinating their populations.

In response to the insurgence of variants B.1.351 (Beta), P.1 (Gamma), and B.1.617.2 (Delta) that have been indicated as variants of concern (VOC), some countries reduced the time interval between vaccine doses. This allowed governments to provide the first dose to a greater number of citizens [54,55]. The planning of an efficient allocation strategy relies on the possibility to simulate the impact of the different choices. In turn, this requires a model of spreading vaccination doses among people [56,57,58].

In [59], the authors present a model for the optimization of vaccination strategies on top of a contact-based simulation model related to Ontario data. The contact strategy is based on the allocation of vaccines according to different age groups. Vaccine distribution and administration are crucial for the healthcare community since they drive the healing phase of any pandemic. We can learn valuable lessons and be prepared for the next global pandemic by analyzing what worked and what went wrong in the disease evolution in relation to the vaccine distribution strategies. Similarly in [60], the authors discuss the problem of promoting vaccines and the comparison of different vaccination strategies. The paper has three relevant results: (i) the choice of optimization strategy depends on the goals (i.e., saving lives or preserving workers); (ii) performances of the simulation are better when more parameters (i.e., social aspects) are considered; and (iii) dynamic strategies (i.e., strategies that may change as a consequence of a modification of external parameters) may outperform the static ones. Such analysis requires the use of accurate models of spreading. Classical SEIR (i.e., Susceptible, Exposed, Infected, Recovered model) models based on differential equations have been recently integrated with contact-based models based on graph theory [61,62,63,64,65]. The use of the SEIR model has been considered as one of the possible alternatives to reduce the diffusion process among countries. Nevertheless, the lack of an adequate analysis of contact among citizens and among security actors and patients has to be considered as a possible cause for the lack of reliable prediction values.

Finally, there have been different proposals on optimal vaccination in different countries, such as South Korea [66], and the WHO’s proposed framework for distributing COVID-19 vaccines among countries [67]. Similarly, strategies depending on different ages (e.g., A Differential Evolution Algorithm Approach [68]) or temporal variation with seasons (e.g., seasonal influenza vaccine under temporal constraints [69]) are also proposed.

## 3. Results

In [70,71], the authors analyze several allocation and distribution strategies for COVID-19 vaccines, and the design of a robust and resilient vaccine distribution network has been defined in [72,73]. Geostatistical modeling has been used to take into account classical logistic constraints (i.e., from the Operations Research domain) and global/local geographical peculiarities, which can greatly vary across different areas (inter-area variability) and within the same area (intra-area variability) [74,75]. Typical constraints and features of this problem, which have to be taken into consideration during the conceptualization and modeling phases, are as follows [76]:The vaccine distribution network should meet geographical constraints, both for populations and vaccination centers (i.e., clinic centres);The locations of the clinics are fixed, while distribution centers may be organized into hubs and spokes;Each nation has a root node of the network, which is the central point of distribution;A hub is refurnished following the WHO guidelines;The coupling of each clinic to the related hub is an a priori choice;The size of the storage device for each center is determined to improve the supply chain management.

Vaccines are distributed through a medical supply chain. In [77], the authors propose a redesign of the vaccine distribution chain with intermediate distribution centers for a number of sub-Saharan countries in Africa. Policies for vaccine delivery may be categorized by considering how much of the the following groups are vaccinated: key workers; clinically vulnerable groups; elderly groups. Existing policies may consider one or many of the previous groups.

The need for an optimal and large-scale vaccine distribution plan for third-world countries has been examined in [78]. The authors describe a clustering-based solution for selecting distribution centers and use a Constraint Satisfaction Problem framework as a support tool for the optimal distribution of vaccines. The efficiency of the proposed models has been demonstrated in India. In [76], the authors analyze the WHO-EPI (Expanded Program on Immunization (EPI)) vaccine distribution for low- and middle-income countries and propose vaccine distribution models. They formulate this as a network design problem [79]. Moreover, a machine learning model is used in [80] to optimize the formulation of the vaccines by improving the binding of peptides. The paper studies the potential impact of vaccination administration on the population by focusing on a diverse set of vaccine peptides. In [81], the authors study how spatial strategies could improve the vaccine distribution efficacy. They focus on designing an optimal vaccination strategy in a country-wide geographic and epidemiological context with high heterogeneity in transmission rate disease history. The simulation is conducted in Italian provinces from January to April 2021 using a public available dataset and a compartmental COVID-19 model tailored to the Italian geographic and epidemiological context.

## 4. Discussion

Vaccination strategies, including validation processes among different countries, strongly influence the pandemic reduction, containment, and solution. The COVID-19 experience should release these models and similar conditions of heterogeneity among nations and the reliability of single vaccine realization. The advantage of a large-scale vaccination campaign is that it rapidly reduces the phenomenon and prevents the virus from learning how to vary and infect the population, even with the first two vaccination doses inoculated.

The COVID-19 pandemic has been thwarted and solved with a return to normal life by mainly applying a two-phase strategy: a containment phase by obliging citizens to respect lockdown rules and then running an immunization procedure vaccination process. The containment has been proven to be an important instance of delaying virus diffusion. Countries with delays in this phase reported increased cases (e.g., see [82]). Regarding results, such a strategy shows a global positive effect. The immunization analysis here is with regard to the possibility of freezing the vaccination strategy as a possible model to be considered and applied in future possible pandemic scenarios. Nevertheless, there have been (even little) differences in the immunization strategies among countries that reported advantages or problems in the containment strategy and in reducing fast virus diffusion. This was also related to the fact that health management rules may vary among different countries [83,84]. The analysis of errors reported during the vaccination processes in terms of applications or strategies related to governments has been studied. For instance, in [85], an analysis of problems and errors related to vaccination procedures is reported. Policy, regulation, and vaccine prioritization among citizens are some issues that must be treated. The strategy of prioritizing older people, as also reported in [51], enabled the saving of both the most number of lives and more years of life. A good prioritization scheme should be part of the strategy, allowing one to choose the best trade-off between saving the maximum number of lives and the most future lives. The implementation and vaccination campaigns regarding healthy or non-healthy citizens, as well as the optimal period to start vaccination on younger persons (under 18 years) or children (under 12 years) is a relevant issue. The parameters used for choice depend on the fragility of older people. Regardless, at the same time, vaccinating teenagers (those in school) is relevant to keep a low diffusion risk (e.g., students sharing the same space for half a day and then returning home, where older people live).

Moreover, errors in vaccination instances have been reported, especially due to large-scale vaccination campaigns; for example, one possible error is with regard to the need for pharma documentation reading as this may contain indications about the second vaccination shot and the risk of overlapping with other drugs used by patients (e.g., heart disease-related ones). Similarly, considering an analysis among different countries, one problem concerned vaccine dose storage and distribution. Also, low errors in vaccination doses have been reported with consequences for patients. Finally, there have been many ethical discussions about the definition of mandatory rules, obliging citizens to participate in vaccination campaigns. Many public roles, as well as a (limited) number of physicians refusing vaccination, could not be maintained by non-vaccinated citizens. The hesitancy in receiving vaccinations has been treated and considered relevant in the strategies (see, for instance, [86]). In [87], hesitancy in receiving vaccinations in 23 different countries was found to be related to mortality and virus diffusion in 2021. This concerns the response of different populations to the vaccination campaign and government rules to convince citizens to be vaccinated. Hesitancy toward vaccination campaigns in young individuals and adolescents has since been dealt with by pediatric associations discussing in favor of universal vaccination [88].

Even if there have been studies regarding the harmful effects of using methods to convince citizens to receive vaccination shots (see [89]), we claim that a vaccination campaign has to be considered as the only exit strategy for a large number of COVID-19 cases. The synchronization and orchestral world large-scale vaccination campaigns allowed us to contain the eventual negative effects of vaccines and variants’ effects.

The immunization process and, most of all, the rapidity of the vaccination process allowed countries to also limit the negative effects of the virus variants. The European Center for Disease Prevention and Control (ECDC) introduced various definitions on SARS-CoV-2 variants, including the so-called variant of concern, as well as variant of interest (VOI), or, finally, variant under monitoring (VUM). Each of these macro-variant definitions are described in terms of: (i) their different features related to the *WHO label*, indicating the official labels used as internationally recognized nomenclature in communications with the public; (ii) *lineage and additional mutations*, which concern characteristics such as the spike protein changes; (iii) the *country* of the virus variation and its detected effects; (iv) *spike mutations of interest*, which include changes to the spike receptor binding domain; (v) *year of first detection*, indicating the first evidence of the variation; (vi) *evidence*, which reports on transmissibility, immunity, infection severity; and (vi) *transmission within European countries*, which describes properties such as: dominant, community, outbreak(s), and sporadic/travel (based on intelligence and direct communication with affected countries). Information regarding variants are strictly related with vaccination campaigns. In particular, variant appearance, diffusion, and geographical area are related to vaccination status. Similarly, the identification of new variants is correlated to the status of the vaccination strategy in the country of interest. The more attention is given to these vaccination campaigns, the more health structures are able to react and identify variant expressions. The evidence regarding transmissibility, severity, and/or immunity (see Table 2) suggests the impact on the epidemiological situation in European countries. In fact, their properties regarding genomic, epidemiological, and in vitro evidence imply moderate confidence. Countries pay attention to the vaccination policies [90], and to the rules, regulations, and restrictions adopted in combination with vaccines. Even if the emergency and lockdown are now a past events, the number of registered COVID-19 cases in the world suggests that the virus has still to be considered with its variants. It is without a doubt that the vaccine campaign helped to solve overcome the worst period. A lot of work, especially in terms of communication and defining social rules, has to be performed to highlight the importance of immunization and of using the right vaccination models for global regulation. It remains the need to prepare for the aftermath of this and for future pandemics through cooperation at a global level [91]. The lessons we learn during the emergency need to be defined as the standard model for immunization protocol, together with the errors, corrections, and good practices that can be summarized from immunization studies. Table 3 reports an example of some of these practices and lessons. Finally, observations about post-pandemic research topics can be found in [92].

### Limitations

The reported analysis is based on the observation of the COVID-19 pandemic evolution. We gathered experiences on virus evolutionary models since 2020 and now report about immunization strategies. Nevertheless, our analysis presents limitations, some of which are reported here:One regards the fact that it is based on studies written mainly during the pandemic, given the short time since the pandemic began. However, it would be helpful to see more research on the long-term effects of vaccination strategies, as well as the impact of different strategies on different populations;A second limitation is that it would be interesting to consider both social and economic factors in a multidisciplinary perspective. For instance, it would be interesting to evaluate how the level of instruction and income impacted the vaccination campaign, as well as the fact that the necessity of starting a fast immunization campaign necessarily lacks the possibility of discussing social impacts;Another limitation is related to the emergence of novel variants. For instance, recombinant variants or Omicron descendants may sensibly change the clinical aspect of the virus and escape current vaccines, which could require the definition of new strategies to contain the diffusion of some of the variants and possible vaccination protocols rebooting.

## 5. Conclusions

COVID-19 vaccination has been the prominent strategy to control the pandemic and to save lives; yet, there is evidence of a lack of sufficient vaccine doses for everyone. Consequently, there is the need for the introduction of an efficient prioritization strategy. The first step is the analysis of different vaccination strategies and outcomes in different countries. In this paper, we shed light on the current scenario regarding vaccines, optimization strategies, and long-term procedures for pandemic monitoring.

In conclusion, we stress the need for studies regarding vaccination processes and virus variants and diffusion by means of geographical modeling to track: (i) the diffusion of the virus, (ii) the status and planning of vaccination supply, and finally, (iii) population movements and habits.

## Figures and Tables

**Table 1 vaccines-11-01496-t001:** Information regarding some of the available vaccines. The table reports information regarding age threshold and authorization timestamp, which allowed the use of a given vaccine with respect to its pharmaceutical producer and country.

Brand	Country	Clinical Trail Status	Age Group	# Shots (Apart)	Time of Achieving Full Vaccination	Date of Approval
Pfizer-BioNTech	USA	Completed	6 months	2 shots	4 weeks after	11 December 2020
				(21 days)	2nd shot	EUA
Spikevax (formerly Moderna)	USA	Completed	6 months	2 shots	2 weeks after	18 December 2020
					(28 days)	2nd shot
Johnson & Johnson’s	USA	Completed	>18	1 shot	2 weeks after	27 February 2021
				(NA)	2nd shot	
AstraZeneca	USA	Completed	NA	NA	NA	February 2021
Novavax	USA	Completed	NA	NA	NA	13 July 2022
COVAXIN	India	Completed	>18	2 shots	2 weeks after	3 November 2021
				(28 days)	2nd shot	
Sputnik V	Russia	Completed	>18	2 shots	2 weeks after	11 August 2020
				(28 days)	2nd shot	
Sputnik Light	Russia	Completed	>18	2 shots	2 weeks after	18 August 2021
				(28 days)	2nd shot	
EpiVacCorona	Russia	Completed	>18	2 shots	2 weeks after	14 October 2020
				(28 days)	2nd shot	
CoviVac	Russia	Completed	>18	2 shots	2 weeks after	February 2021
				(28 days)	2nd shot	
Sinopharm-BBIBP	China	Completed	>3	2 shots	2 weeks after	30 December 2020
				(28 days)	2nd shot	
CoronaVac	China	Completed	>18	2 shots	2 weeks after	1 June 2021
				(28 days)	2nd shot	
Convidecia	China	Completed	>18	2 shots	2 weeks after	4 September 2022
				(28 days)	2nd shot	
Sinopharm-WIBP	China	Completed	>18	2 shots	2 weeks after	25 February 2021
				(28 days)	2nd shot	
RBD-Dimer	China	Completed	>18	2 shots	2 weeks after	1 December 2021
				(28 days)	2nd shot	
Minhai	China	Completed	NA	NA	NA	14 May 2021
QazCovid-in	Kazakhstan	Completed	NA	NA	NA	April 2021
Zydus Cadilla	India	Completed	>18	1	1	1 July 2022
Covishield	India	Completed	>18	1	1	1 January 2021
Incovac BBV154	India	Completed	>18	1	1	24 December 2022
COVOVAX	India	Completed	>18	1	1	1 November 2021
HGCO19	India	Phase III Trial	NA	NA	NA	NA

**Table 2 vaccines-11-01496-t002:** SARS-CoV-2 variants of concern (VOC) (x: A67V, Δ69-70, T95I, G142D, Δ143-145, Δ211, ins214EPE, G339D, S371L, S373P, S375F, K417N, N440K, G446S, S477N, T478K, E484A, Q493R, G496S, Q498R, N501Y, Y505H, T547K, D614G, H655Y, N679K, P681H, N764K, D796Y, N856K, Q954H, N969K, L981F). Table taken from [93].

WHO Label	Lineage + Additional Mutations	Country First Detected (Community)	Spike Mutations of Interest
**Year and Month First Detected Transmission in EU/EEA**	**Evidence for Impact on Transmissibility**	**Evidence for Impact on Immunity**	**Evidence for Impact on Severity**
Beta	B.1.351	South Africa	K417N, E484K, N501Y, D614G, A701V
September 2020 Community	Yes	Yes	Yes
Gamma	P.1	Brazil	K417T, E484K, N501Y, D614G, H655Y
December 2020 Community	Yes	Yes	Yes
Delta	B.1.617.2	India	L452R, T478K, D614G, P681R
December 2020 Dominant	Yes	Yes	Yes
Omicron	B.1.1.529	South Africa and Botswana	(x)
November 2021	Yes		Sporadic/Travel

**Table 3 vaccines-11-01496-t003:** Examples of good practices resulting from analyzing vaccinations procedures.

Good Practice	Effects	Notes
Design WHO-compliant strategies	Possibility to learn from other countries to implement effective campaigns that are adapted to the country context	Positive effects related to uniformly applied immunization strategies
Involve general medical doctors	General medical doctors are aware of the local context and patients have a high level of trust in them, thus they may efficiently quicken the campaign process	Medical family doctors are the ones maintaining the timeline of citizen diseases, and thus will play an important role and achieve positive effects
Implement social communication strategies to involve world-scale populations	Social engagement is highly relevant for the success of vaccination; thus, the design of ad hoc social media campaigns has increased the number of vaccinations	Possible negative effects include the reliability on false information (fake news)
Dynamic strategies	By continuously monitoring the evolution of the campaign, it is possible to adapt the approach to maximize results	Possibility to generate distrust in the strategy
Integrate COVID-19 vaccination monitoring and information systems into the primary healthcare information systems	This may change the perception of COVID-19 campaigns, from an exceptional pandemic to a normal disease like other infectious diseases such as HIV	This may increase the strength of the vaccination campaign as a classical healthcare routine
Improve electronic data collection and sharing through the use of electronic tools for monitoring	The progressive transition from paper-based campaigns to a digital system is essential to allow all actors to have real-time data in response to the speed of the spreading	Improves the speed of the response to the pandemic

## Data Availability

Data sharing not applicable. No new data were created or analyzed in this study.

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
