# Peer review of "Strategies and Trends in COVID-19 Vaccination Delivery: What We Learn and What We May Use for the Future"

_vaccines, 2023, doi:10.3390/vaccines11091496_

Round 1

Reviewer 1 Report

I think this article will be of interest to Vaccines Journal readers, the topic is quite important for the scientific community. The strategy and distribution of Covid-19 vaccines is still of a high interest, even during this endemic stage.

In addition, the article is well written so congrats to the authors on this. As suggestions for accepting this publication I'd recommend the following:

- On table 1, there are many covid-19 vaccines listed but I don't see some younger peds indications (below to 12yrs), I'm not sure why these were not included (e.g. Pfizer or Moderna). In addition, I'd suggest to include a column with the year of regulatory approval in their own country, that will provide more information

- I saw some critics to the SEIR models, that should be better justified by authors, why you think those models are not accurate and provide examples, that will be beneficial

- I haven't seen a good discussion within the manuscript on the pros and cons of the immunization strategies, probably taking examples what went well and what went wrong and using country cases would be interesting to have, I see the discussion section of the manuscript too short and it could gain more insights with these type of discussion. The same with the distribution of covid-19 vaccines, very few had been said on the successful and unsuccessful cases with distribution, since it is part of the objectives from the manuscript I was expecting some more examples, probably you may use tables to highlight some good practices that could be shared on this topic

- Currently the article is lacking of the limitations of the analysis and the literature review done by the authors, I'd suggest to consider this as well in the discussion section either

Author Response

Reviewer 1: I think this article will be of interest to Vaccines Journal readers, the topic is quite important for the scientific community. The strategy and distribution of Covid-19 vaccines is still of a high interest, even during this endemic stage.

In addition, the article is well written so congrats to the authors on this. As suggestions for accepting this publication I'd recommend the following:

- On table 1, there are many covid-19 vaccines listed but I don't see some younger peds indications (below to 12yrs), I'm not sure why these were not included (e.g. Pfizer or Moderna). 

In addition, I'd suggest to include a column with the year of regulatory approval in their own country, that will provide more information

Answer: Firstly let us thank the reviewer for comments and considering the effort. Indeed this study has been proposed to analyze vaccination processes on a large scale that could be considered as a reference to contrast possible new pandemic events. Many papers have been published designing the pandemic events, little in terms of endemic current phenomena. 

Following the reviewer’s indications we updated Table 1 and we inserted information about vaccination protocols for under 12. We also studied and added information about approval intervals distinguished by country as also required by Reviewer 2. This is also a relevant parameter that has to be considered in the study to cover the topic regarding strategy applications variability depending on countries regulation. We thank both reviewers for the comments. We updated the table description and we also added few lines of comments (please see bold font in the new version). 

- I saw some critics to the SEIR models, that should be better justified by authors, why you think those models are not accurate and provide examples, that will be beneficial

Answer: We apologize for the lack of clarity of the sentence. We modified the sentence to better explain that classical SEIR models may gain benefit from the integration with contact based models, thus that a greater control can be used by means of contact based models such as oriented graphs with its scalable managing process. 

- I haven't seen a good discussion within the manuscript on the pros and cons of the immunization strategies, probably taking examples what went well and what went wrong and using country cases would be interesting to have, I see the discussion section of the manuscript too short and it could gain more insights with these type of discussion. 

Answer: We thank the reviewer for such a comment. We improved the discussion section reporting comments about pros and cons of the immunization process. We think that the discussion has been enriched guiding readers through the message of information regarding immunization strategies. 

We restructured the discussion section also according to the editor's indication about the necessity of rewriting part of it. One of the main messages contained in such a contribution regards the possibility of freezing the vaccination strategy as a possible model to be considered and applied in future possible pandemic scenarios. According to the reviewer observations, applications of immunization strategy may vary among countries with different health management rules. This reported advantages or problems in containment strategy as well as in reducing fast virus diffusion. 

-I was expecting some more examples, probably you may use tables to highlight some good practices that could be shared on this topic.

Answer: We thank the reviewer for such a comment. We enriched the discussion by adding some additional examples and references to good practices reported in a table. 

- Currently the article is lacking of the limitations of the analysis and the literature review done by the authors, I'd suggest to consider this as well in the discussion section either

Answer: We thank you reviewer for such an observation. We rewrote and restructured Discussion section and we added a paragraph “Limitations” 

Reviewer 2 Report

Comments related to the article

·         Page no 1 (line no 29)- On 1 January 2021, the Drug Controller General of India (DCGI) approved emergency use of the Oxford–AstraZeneca vaccine (local trade name "Covishield") so the first Indian vaccine for covid 19 was released on 1st january 2021 that is covishield. (not the end of the year as in the text)

References-

o   BBC News. 3 January 2021. Retrieved 22 April 2021.

o   The Guardian. Archived from the original on 24 March 2021. Retrieved 22 April 2021.

o   BBC News. 6 April 2021. Retrieved 22 April 2021

·         Page no 2 – (line no 37) Atleast 72.3 percent of world population has received single dose of covid vaccine by the end of 2022.

Reference – New York Times posted on March 13th 2023.

·         Page no 4 in the Table 1 (heading)- An Extract of available Vaccines.

Table should also include more vaccines produced in India for covid 19

1.      Zydus Cadilla vaccine -plasmid DNA vaccine

2.      Covishield

3.      BBV154 - Intranasal Vaccine- Bharat Biotech is conducting Multicenter Study to Evaluate the Reactogenicity, Safety, and Immunogenicity of an Intranasal Adenoviral vector COVID-19 vaccine (BBV154) in Healthy Volunteers. BBV154 is an intranasal vaccine stimulates a broad immune response – neutralizing IgG, mucosal IgA, and T cell responses.  Immune responses at the site of infection (in the nasal mucosa) – essential for blocking both infection and transmission of COVID-19.

4.      COVOVAX- Indian Council of Medical Research and Serum Institute of India jointly performing a phase 2/3, observer-blind, randomized, controlled study to determine the safety and immunogenicity of COVOVAX [SARS-CoV-2 recombinant spike protein nanoparticle vaccine (SARS-CoV-2 rS) with Matrix-M1™ adjuvant] in Indian adults

5.      mRNA based vaccine (HGCO19)-Randomized, Phase I/II, Placebo-controlled, Dose-Ranging, study to evaluate the Safety, Tolerability and Immunogenicity of the candidate HGCO19 (COVID-19 vaccine) in healthy adult subjects. The trial is being conducted by Gennova Biopharmaceuticals Limited, Pune.

Reference- ICMR 2023

·         Page no 5 (line no 150)- India launched its vaccination programme for covid 19 on 16th January 2021

Author Response

 Reviewer 2 :Comments related to the article

  •     Page no 1 (line no 29)- On 1 January 2021, the Drug Controller General of India (DCGI) approved emergency use of the Oxford–AstraZeneca vaccine (local trade name "Covishield") so the first Indian vaccine for covid 19 was released on 1st january 2021 that is covishield. (not the end of the year as in the text) References 

-   BBC News. 3 January 2021. Retrieved 22 April 2021.

-   The Guardian. Archived from the original on 24 March 2021. Retrieved 22 April 2021.

-   BBC News. 6 April 2021. Retrieved 22 April 2021

Answer: We thank the reviewer for such useful comments and very useful information. We modified the text accordingly and we removed the reference to the end of 2021. We also added as a matter of example the fact that in some countries as India or Italy the immunization started earlier wrt other countries. 

Page no 2 – (line no 37) Atleast 72.3 percent of world population has received single dose of covid vaccine by the end of 2022.

Reference – New York Times posted on March 13th 2023.

Answer: We modified the text accordingly and we apologize for missing information. 

Page no 4 in the Table 1 (heading)- An Extract of available Vaccines.

 Table should also include more vaccines produced in India for covid 19

  1.     Zydus Cadilla vaccine -plasmid DNA vaccine
  2.     Covishield
  3.     BBV154 - Intranasal Vaccine- Bharat Biotech is conducting Multicenter Study to Evaluate the Reactogenicity, Safety, and Immunogenicity of an Intranasal Adenoviral vector COVID-19 vaccine (BBV154) in Healthy Volunteers. BBV154 is an intranasal vaccine stimulates a broad immune response – neutralizing IgG, mucosal IgA, and T cell responses.  Immune responses at the site of infection (in the nasal mucosa) – essential for blocking both infection and transmission of COVID-19.
  4.     COVOVAX- Indian Council of Medical Research and Serum Institute of India jointly performing a phase 2/3, observer-blind, randomized, controlled study to determine the safety and immunogenicity of COVOVAX [SARS-CoV-2 recombinant spike protein nanoparticle vaccine (SARS-CoV-2 rS) with Matrix-M1™ adjuvant] in Indian adults
  5.     mRNA based vaccine (HGCO19)-Randomized, Phase I/II, Placebo-controlled, Dose-Ranging, study to evaluate the Safety, Tolerability and Immunogenicity of the candidate HGCO19 (COVID-19 vaccine) in healthy adult subjects. The trial is being conducted by Gennova Biopharmaceuticals Limited, Pune. Reference- ICMR 2023

Answer: We thank you for the very useful comment. We updated the table and we inserted the list of vaccines enriching the manuscript.

  •         Page no 5 (line no 150)- India launched its vaccination programme for covid 19 on 16th January 2021

Answer: We thank you for the very useful comment. We added such information enriching the information regarding the vaccination program.

Round 2

Reviewer 1 Report

I'd agree with this version for publication purposes.